# Towards 6G IoT: Tracing Mobile Sensor Nodes with Deep Learning Clustering in UAV Networks

**DOI:** 10.3390/s21113936

**Published:** 2021-06-07

**Authors:** Yannis Spyridis, Thomas Lagkas, Panagiotis Sarigiannidis, Vasileios Argyriou, Antonios Sarigiannidis, George Eleftherakis, Jie Zhang

**Affiliations:** 1Department of Electronic and Electrical Engineering, The University of Sheffield, Sheffield S1 3JD, UK; ispyridis1@sheffield.ac.uk (Y.S.); jie.zhang@sheffield.ac.uk (J.Z.); 2Department of Computer Science, International Hellenic University, 654 04 Kavala Campus, Greece; tlagkas@cs.ihu.gr; 3Department of Electrical and Computer Engineering, University of Western Macedonia, 501 31 Kozani, Greece; 4Department of Networks and Digital Media, Kingston University, London KT1 1LQ, UK; Vasileios.Argyriou@kingston.ac.uk; 5Sidroco Holdings Ltd, Nicosia 1077, Cyprus; asarigia@sidroco.com; 6Computer Science Department, CITY College, University of York Europe Campus, 546 26 Thessaloniki, Greece; g.eleftherakis@sheffield.ac.uk

**Keywords:** unmanned aerial vehicles, deep learning, IoT, 6G, graph convolutional network, sensor tracking, RSSI

## Abstract

Unmanned aerial vehicles (UAVs) in the role of flying anchor nodes have been proposed to assist the localisation of terrestrial Internet of Things (IoT) sensors and provide relay services in the context of the upcoming 6G networks. This paper considered the objective of tracing a mobile IoT device of unknown location, using a group of UAVs that were equipped with received signal strength indicator (RSSI) sensors. The UAVs employed measurements of the target’s radio frequency (RF) signal power to approach the target as quickly as possible. A deep learning model performed clustering in the UAV network at regular intervals, based on a graph convolutional network (GCN) architecture, which utilised information about the RSSI and the UAV positions. The number of clusters was determined dynamically at each instant using a heuristic method, and the partitions were determined by optimising an RSSI loss function. The proposed algorithm retained the clusters that approached the RF source more effectively, removing the rest of the UAVs, which returned to the base. Simulation experiments demonstrated the improvement of this method compared to a previous deterministic approach, in terms of the time required to reach the target and the total distance covered by the UAVs.

## 1. Introduction

The advent of unmanned aerial vehicles (UAVs) has led to their recurrent deployment as aerial communication platforms in wireless networks. Research in the domain of relay networks has experienced a growing UAV initiative, with the incorporation of UAVs providing a versatile and appealing solution, since they provide increased coverage, while offering improved connectivity to devices on the ground. Courtesy of their high cost-effectiveness and flexible deployment prospects, UAVs have a presiding effect in current realisations of Fifth-Generation (5G) cellular networks [1], conceptualising demanding networking techniques and covering all communication prerequisites [2]. In the near future, it is expected that the era of UAV-assisted networks will emerge, through the deployment of countless UAVs in every layer of human activities, enabling several smart applications in the process [3].

The rapidly growing paradigm of the Internet of Things (IoT) is also an area in which UAVs will play a dominant role [4]. Focusing on the incorporation of state-of-the-art technologies, the IoT describes a connection of several smart objects, which exchange data over the Internet, targeting the enhancement of daily life. Devices in the IoT include sensors that are often constrained by a limited battery and thus usually broadcast at short ranges. In this context, UAVs can be employed as data-collection stations, which determine the device’s location being out of transmission range from the destination and afterwards establish a connection with the latter in order to re-transmit the collected data [5,6]. UAVs in the role of aerial stations are expected to remarkably enhance IoT infrastructures, through the dynamic provision of stable device connections, as opposed to terrestrial base stations which may often be characterised by a lack of line-of-sight (LoS). As a result, using UAVs can avoid shadowing effects in the wireless channel by ensuring LoS in the interconnections, providing a cost-effective solution for serving ground devices in an efficient way [7].

With the current deployment of 5G networks being established globally, the research community has already started to investigate ways to go beyond the 5G capabilities and pave the way toward the Sixth-Generation (6G) of wireless communication systems. Although precise requirements have not yet been defined, 6G architectures are expected to offer wireless universal connectivity worldwide, targeting smart applications that require bitrates on the scale of terabits per second, while achieving a sub-millisecond latency [8,9]. The paradigm of UAV-assisted localisation has recently started to emerge in 6G research, in which UAVs assume the role of anchor nodes (ANs) and are used as a reference to determine the position of ground devices [10]. Due to their high flying altitude, UAVs have a higher probability of establishing LoS connections with terrestrial nodes, offering localisation services when ground ANs are unavailable or unable to deliver reliable connections. Following this paradigm, 6G technologies have the ambition of seamlessly integrating ground and aerial wireless networks through the use of UAVs, establishing ubiquitous connectivity in the process [11].

At the same time, ever since the introduction of machine learning (ML) as a key artificial intelligence (AI) approach, extensive research has been conducted towards its intensive adoption in wireless networks, aiming to provide computationally efficient and robust algorithms [12]. In essence, ML focuses on the development of models that target knowledge acquisition problems, by utilising learning procedures that improve the capabilities of the corresponding systems through the detection of patterns in the provided data [13,14]. The application of ML techniques in wireless sensor networks can offer substantial benefits, mainly due to the dynamic nature of the environments that these networks are used to monitor. For instance, nodes in such a network may be mobile, or even if stationary, their location may differ over time due to changes in the environment (e.g., soil erosion, sea turbulence). Another example comes from the smart city paradigm, where AI techniques are applied to enhance transport in the new mobility era [15]. In addition, wireless networks may be used for collecting data in hardly accessible or hazardous locations, often characterised by unpredictable conditions, or in environments where accurate mathematical models cannot be developed [16]. Robust systems relying on ML can be utilised in these situations, enabling the network to reconfigure itself or to adapt accordingly, utilising low-complexity approximations. As a result, ML approaches are expected to be the key enabling technology for the efficient operation of wireless networks, supporting several novel applications in this context.

Given the energy-constrained nature of IoT nodes, UAVs can help minimise the transmission power required by target sensors by dynamically adjusting their altitude, while at the same time ensuring robust coverage. As a result, they offer a versatile and efficient solution towards locating sensors and providing relay services to remote IoT infrastructures in the context of 6G. The approach presented in this paper targets such an application and enables a group of UAVs to reach an IoT sensor node located in an unknown position, by relying only on the radio frequency (RF) broadcast signal emitted by the latter. The system utilises a deep learning (DL) model, which performs clusterisation in the UAVs based on their received signal strength indicator (RSSI) and their current location. At regular intervals, the proposed algorithm selects distant groups of UAVs based on the model’s outcome, returning them to the base in order to save energy and to keep the clusters active that seem to have approached the target more effectively. The key contributions of this work can be summarised as follows:We introduced a graph representation of the UAV network that enabled the utilisation of graph convolutional network (GCN) architectures;The clusterisation was performed in the network at regular intervals based on the RSSI and the current locations of the UAVs, with a dynamic number of clusters at each phase;We utilised the optimisation of an RSSI deep learning loss function to determine the final clusters;The introduced method allowed the UAVs to trace the target sensor node without explicit knowledge of its location or the use of distance estimates.

Simulation experiments demonstrated the improvement of this method compared to our previous deterministic approach [17] in terms of time efficiency when approaching the target, but also in terms of distance covered by the UAVs, therefore denoting increased energy efficiency as well.

The rest of the paper is structured as follows: Section 2 provides an overview of the state-of-the-art in the localisation of RF sources. Section 3 presents the proposed DL approach and describes the tracing algorithm. Section 4 discusses the test-bed devised to evaluate the introduced method and outlines the considered scenarios, while Section 5 presents the results obtained through the evaluation experiments. Finally, Section 6 provides concluding remarks.

## 2. Related Work

A common approach for the localisation of unknown nodes in wireless networks involves the employment of ANs in tactical locations, aware of their position usually through a Global Navigation Satellite System (GNSS) or through pre-defined information and manual programming. Using the known locations of the ANs, the process of trilateration or triangulation is traditionally used, relying on distance estimates from the unknown sensor to approximate its position. In light of the inconsistent accuracy of the measurements caused by signal fluctuations, estimation methods have more recently shifted towards optimisation techniques in order to enhance the RSSI precision [18].

One of the key challenges that arises in RF source localisation using UAVs is the optimal trajectory control in order to increase the accuracy of the target’s position estimates and obtain high information measurements [19]. Several methods for solving this problem involve model predictive control (MPC) algorithms, also referred to as receding horizon control (RHC), for the path planning of the UAVs. The authors in [20] proposed a hierarchical MPC algorithm to control the trajectory of the UAVs with the goal of achieving the required target estimation accuracy in the shortest time possible. Carrying direction-of-arrival (DOA) sensors, the involved fixed-wing UAVs were tasked to detect and track the RF source, which was either stationary or moving with a small velocity. By incorporating the hierarchical variant in the MPC, the algorithm allowed optimising for long-term goals with simulation experiments, demonstrating its effectiveness.

Relying on DOA sensors is not always convenient, especially when involving small multi-rotor UAVs. As a result, simpler alternative methods such as passive RSSI sensors, have been suggested for use in real-world applications. In this context, an RHC algorithm governing the path planning of an RSSI sensor-equipped UAV swarm was proposed in [21], towards the goal of localising a mobile RF transmitter. The UAVs determined the target position estimate in cooperation, using an extended Kalman filter (EKF), and calculated the optimal path, by maximising the determinant of the Fisher information matrix (FIM) of the potential set of actions. The objective of multiple target tracking was achieved by the RHC algorithm proposed in [22], which was distributable across multiple agents. Relying on ergodic theory to study the search trajectory, the introduced technique utilised hybrid systems theory to calculate control actions that improved the ergodicity in an MPC manner. The authors demonstrated both in a simulation environment using UAVs and in real-world experiments using a commercial robot that the agents could reliably track the targets in real time based on bearing-only measurements.

In the work presented in [23], the authors considered a scenario where the RF source was only intermittently transmitting, in which case the RSSI measurements were not continuously available. In the target application, the UAVs were set to patrol an area of interest and had to locate the intermittent RF source, based on a two-stage technique. The first stage involved the task of localisation, where the UAVs estimated the target location, given the previous RSSI measurements, based on a recursive Bayesian estimator. In the second stage, the optimal future trajectory was determined, with the goal of reducing the localisation error by taking into consideration the current estimation, using a steepest descent path-planning algorithm. Simulation results demonstrated that the recursive Bayesian estimator slightly outperformed an EKF in the first stage, while the steepest descent algorithm displayed major benefits in the second stage, when compared to a bioinspired approach.

The scenario of localising intermittently transmitting mobile RF sources was also examined in [24], using fixed-wing UAVs equipped with angle-of-arrival (AOA) sensors. The proposed system relied on a decentralised architecture whereby each UAV determined its path cooperatively based on the information received by neighbouring UAVs and on the data provided by its sensor. Control actions were taken based on a cost function regarding the distance that each UAV needed to cover to assist with the localisation, the number of active UAVs assisting in the current target location and the number of adjacent UAVs. In the case that the cost function went below zero, the corresponding UAV proceeded to assist in the localisation; otherwise, it retained its current actions. To accommodate the intermittent transmission, an efficient path-planning algorithm was proposed that governed the revisiting process at every location. The authors investigated the performance of three localisation techniques—triangulation, angle-rate and Kalman filtering—using simulation experiments and concluded that a combination of these could yield a generally accurate localisation method.

Employing the measurements of the Doppler frequency in the received signal, the method in [25] managed to lead a UAV to a distant RF emitter, by continuously adjusting the trajectory using bearing estimates. At a first stage, the UAV followed a random circular trajectory, while storing frequency and bearing measurements. In order to filter the stored values and minimise the effects of multipath fading, an outlier rejection technique was applied to the frequency measurements. In the second stage, the UAV derived changes in the Doppler frequency at regular intervals for a persistent trajectory control that led it towards the RF source. The authors demonstrated the increased performance of this method through simulation experiments; however, the required receiver operations were more complex, when compared to simply extracting the RSSI.

The localisation of mobile phones through an ML approach was presented in [26]. Relying on the RF signal of intermittent probe requests sent by the devices on the ground, a group of UAVs was used to capture these request messages from different locations and estimate the phone’s position based on the RSSI. In order to classify the location zone where the device belonged, the model had to be initially trained on the target area, using a random forest algorithm, and then, the method was applied to locate the target zone with a high accuracy.

The method proposed in this work differed from the aforementioned studies, in that it aimed to cover critical aspects associated with real-world scenarios. Towards this goal, it utilised simple RSSI sensors equipped on the UAVs, as opposed to more complex antennas, such as DOA or AOA sensors, that have been proposed in other approaches. In addition, it incorporated the use of small and flexible multi-rotor UAVs, in contrast to less versatile and cost-effective fixed-wing UAVs that were often used in similar works. When it comes to the tracking problem, the introduced method diverged from related studies, since it did not involve the conversion of the received signal’s power to distance from the target, which would inevitably have errors due to signal fluctuations. Finally, the deep learning model that was proposed did not require pre-training on a specific scenario to provide the solution, but operated in an unsupervised manner.

## 3. Methodology

In the considered problem, the target node refers to an IoT gateway, which might be located on or near an IoT device. The involved UAVs act as mobile relays, aiming to collect measurements or status updates from the target sensor. These measurements have as the destination a 6G cellular base station, which will receive them through the assistance of the UAVs. The aim was to trace the IoT source node by exploiting wireless signal observations, through omni-directional RSSI sensors equipped on the UAVs.

### 3.1. Tracing Algorithm

The goal of the proposed tracing algorithm was to lead the group of UAVs to the mobile sensor’s unknown location, as quickly as possible. Since the UAVs are only equipped with RSSI sensors, they have to rely solely on knowledge about the signal strength measured at their antenna for the whole process. In short, the process relies on differences in RSSI measurements over time, whereby increases in signal strength indicate increasing proximity to the target, while decreases signify diversion. Instead of relying on the free-space path loss, the proposed scheme relies on a more accurate model for the signal attenuation in moving networks, as measured experimentally in [27] and comprehensively described in [17]. Utilising that model, the RSSI at each UAV is calculated by the following formula:(1)$$RSSI=P_{x}-\left[41.1log_{10}\left(d\right)+17.2+20log_{10}\left(f/5\right)\right]+G_{tx}+G_{rx},$$
where $P_{x}$ is the transmission power of the target sensor in dBm, the term inside the square brackets is the path loss, where *d* is the distance between the transceiver and receiver, *f* is the signal frequency and $G_{tx}$ and $G_{rx}$ represent the transceiver’s and receiver’s antenna gain, respectively.

In the considered scenario, all UAVs were deployed in a common location and started moving towards random, constant directions, until a predefined threshold of the RSSI difference between the two closest UAVs was measured. At regular intervals, the GCN model performed clustering in the network and determined groups of UAVs based on their proximity to the target and their corresponding locations. Following an unsupervised approach, the model was able to divide the UAVs into appropriate groups, without requiring any prior training at any given location. The algorithm then ordered the clusters according to the target proximity and removed those that appeared distant, with the associated UAVs returning back to the base. Following this process, the model was able to retain the groups of UAVs that seemed to approach the target more effectively, while minimising the total energy expended by landing the unnecessary UAVs. The whole tracing process is demonstrated in the flowchart of Figure 1. When the RSSI termination threshold was reached, the remaining group switched to a cooperative mode, and the UAVs navigated based on the distributed approach presented in [17], able to trail the sensor at a close distance. The optimal interval at which the clustering process occurred was examined through simulation evaluations, which determined that in general, lower intervals yielded better performance. For the final implementation, the interval of 30 s was selected, since it provided an adequate performance, while being appropriate for use in a realistic scenario.

### 3.2. Deep Learning Clustering for the Determination of UAV Groups

Deep learning methods can be used to offer powerful solutions in graph partitioning problems, by utilising the implicit architecture of GCNs. The approach followed in this work combined the GAP framework for graph partition [28], with a deep learning loss function of the RSSI, aiming to group the UAVs into appropriate clusters based on their proximity to the target. This method was selected as it was capable of generalisation and allowed the joint optimisation for the target loss function, leading to the fine-tuning of the model for this specific graph structure. Following this process, the algorithm ordered the clusters and retained those that achieved the highest proximity.

Leveraging the capabilities of GCNs, the devised model relied on four such layers to perform convolution on the graph data, each followed by a corresponding ReLU layer. The architecture was completed by utilising two linear layers to scale the output to the selected number of clusters, followed by a softmax output layer in the end, as demonstrated in Figure 2. This specific setup allowed for better feature representations and correlations among the input parameters in the considered clustering problem. The size of the network was selected according to the involved data, aiming to avoid over-fitting issues. In order to provide the clustering prediction, the DL model required the graph representation of the UAV network.

#### 3.2.1. Graph Representation of the Network

To effectively make use of the GCN layers and provide the required input to the model, the system had to be defined using a graph representation. In the considered network, all nodes interchanged information with each other, and thus, a fully connected graph was constructed, representing the UAVs as vertices, while the edges denoted the interconnections among the UAVs. Using this representation, the degree and adjacency matrices were derived, allowing the deduction of the normalised Laplacian matrix of the graph, as follows:(2)$$L_{normalised}=I-D^{-\frac{1}{2}}AD^{-\frac{1}{2}},$$
where **I** is the identity matrix and **D** and **A** are the degree and adjacency matrices, respectively. In addition, a new matrix was created, holding information about each UAV pertaining to the location coordinates and the RSSI from the target. Using the above matrices, the final GCN model was constructed and trained to provide the optimal clusters.

#### 3.2.2. Deep Learning Loss Formulation

The training process utilised the following loss function that aimed to maximise the RSSI at each cluster:(3)$$L_{RSSI}=\frac{1}{N}\sum_{n=1}^{N}\left|Yr\right|,$$
where **Y** is the output matrix from the neural network and **r** is a vector containing the RSSI values. By optimising this loss function, the deep learning model determined the partitions and where each node belonged. To ensure that the number of UAVs was balanced in each cluster, an additional loss function was used:(4)$$L_{balance}=max\left(0,\frac{n}{k}-\sigma 1^{T}Y\right),$$
where *n* is the number of UAVs, *k* is the number of clusters and σ is a hyperparameter for adjusting the relief degree.

#### 3.2.3. Determining Optimal Number of Clusters

The nature of the introduced tracing algorithm was such that the deep learning model was called with different numbers of UAVs, located at varying positions each time. Therefore, it became inherently arduous to decide the number of desired clusters the model ought to produce in advance. Since this number affected the performance of the algorithm, it was crucial that it be defined dynamically, according to the imminent situation. Even though the intuition suggests that a higher number of clusters would lead to a better fit, there was a point where diminishing returns stopped justifying the additional cost and the phenomenon of over-fitting appeared. This point can be deduced using several heuristic methods, such as the “elbow” method, which was chosen in this work.

The performance of the proposed algorithm varied for different numbers of clusters, but because in each clustering interval, the setup was unique for every experiment, it was intrinsically difficult to deduce a universal correlation. For this reason, the “elbow” method appeared exceptionally useful, since it helped to automate the whole process and determine the appropriate number of clusters. Even though in certain cases, it might yield a suboptimal solution, on most occasions, the result was optimal.

The concept behind the “elbow” method is to perform *k*-means clustering for a range of *k* values, calculating the sum of squared errors (SSE) for each *k*. After plotting the SSE at each value of *k*, the corresponding line chart appeared as an arm, as depicted in Figure 3, where the elbow was the optimal *k* value, after which the inclusion of an additional cluster did not result in significantly better modelling of the dataset. Mathematically, the “elbow” can be deduced by calculating the gradient of the graph between each pair of *k* and determining where the highest change appears.

## 4. Simulation Setup

A test-bed comprised of two tracing scenarios was developed in order to evaluate the performance of the proposed algorithm:In the first scenario, the speed of the UAVs was set to 40 km/h, and the target sensor was placed in a random position, 2 km away from the UAV deployment location;In the second scenario, the target was positioned 3 km away from the deployment location, and the UAVs’ velocity was set to 50 km/h.

In both cases, the target sensor’s movement was based on a random waypoint model, and the UAVs executed their flying routes using the proposed algorithm, as explained in Section 3. Each experimental scenario was executed for different values of the standard deviation of noise due to slow fading (σ) to examine the performance of the algorithm in various conditions. The effect of fast fading was considered negligible since the involved distances were large and the UAVs flew in an open environment with not many obstacles that could result in multi-path propagation. In addition, the RSSI threshold at which the clustering process was terminated was also examined. Finally, every experiment was repeated 15 times, using a different random seed for the target location and the UAV directions each time, and the average result was used to determine the outcome with higher confidence. The simulation parameters, including information about the sensor and the UAVs, are outlined in Table 1.

The experiments were conducted using the simulator that was built in our previous work, which was updated to integrate the developed DL model and obtain the clustering result at each interval using the new algorithm. In the considered scenarios, the simulator was set to export a file containing all the UAV positions and RSSI measurements every 30 s. The DL model used the exported file as the input and returned a set of integers, indicating the indices of the UAVs corresponding to the cluster that had to be removed. In the first execution, the input size of the DL model was 45 × 3, since there were initially 45 UAVs each with a unique RSSI measurement, as well as *X* and *Y* coordinates. The *Z* coordinate was not required, as it corresponded to the flying altitude, which was the same for all UAVs and therefore did not affect the clustering outcome. For subsequent executions, the input size depended on the number of remaining UAVs. While conducting the experiments, the simulator provided graphical representations of the simulated entities, as well as numerical information regarding the parameters and current metrics, as depicted in Figure 4. The final results were exported to the corresponding spreadsheets after all the experiment finished.

## 5. Evaluation

This section presents the results of the experiments conducted for each of the two scenarios. The key objective of the simulations was to determine whether our previous deterministic approach could be efficiently replaced by the new clustering-based DL method and offer a thorough evaluation of the two algorithms in various conditions. The performance was quantified in terms of the time required for the UAV group to reach the target sensor, the average distance covered per UAV throughout the tracing process and, finally, the total distance covered by all UAVs.

### 5.1. Results

Figure 5 and Figure 6 illustrate the time required in seconds for the sensor to be reached, versus the standard deviation σ of the additive Gaussian noise, when the target was positioned at a distance of 2 km and 3 km, respectively. The two plots depict the results of the new method at different RSSI termination thresholds and the previous algorithm [17]. In the first case, the new method tended to perform best at a threshold of 20 dB, but on average, a termination threshold of 15 dB seemed to be a better overall choice for both scenarios. Nevertheless, the new method was able to outperform the previous approach consistently at all thresholds, with a gained advantage of almost 60 s on some occasions.

Figure 7 and Figure 8 similarly present the results of the two methods in terms of the average distance covered per UAV and the total distance covered by all UAVs, respectively, when the target’s initial distance was 2 km, while Figure 9 and Figure 10 depict the same results for the target at an initial distance of 3 km. Overall, the new method displayed the highest efficiency at a threshold of 20 dB, with each UAV covering on average approximately 1 km less when compared to the previous approach. The advantage of the new method became dominant when examining the total distance covered by all UAVs throughout the tracing process, as demonstrated in Figure 8 and Figure 10. This was expected, since the new algorithm removed distant groups of UAVs at each clustering interval, keeping only the clusters that displayed the highest effectiveness in approaching the target. In addition, observing the values in these plots offered an indication of the number of UAVs that remained active, with lower total distance implying fewer UAVs remaining in the end.

Table 2 offers an overview of the average results in the first scenario, for all values of the standard deviation σ of the additive noise, both for the new DL method and the previous approach, at all RSSI termination thresholds. The new method demonstrated higher efficiency in all metrics, irrespective of the threshold, especially at reducing the total distance covered by the UAVs. The results also revealed that the new method was on average more efficient both in terms of the time required to reach the target and the total distance covered when the threshold of 20 dB was selected. As a result, this choice would be appropriate for a potential real deployment of this tracing scheme, managing to outperform all other approaches and at the same time expending less total energy to do so, by requiring a much shorter distance to be covered by the UAVs.

### 5.2. Ablation Study

To gain insight regarding the benefits of using the “elbow” method to determine the optimal number of clusters dynamically, instead of arbitrarily setting a desired number, the results in the first scenario without a dynamic cluster quantity were also examined. Figure 11 depicts the time required to reach the target in each case, when the RSSI termination threshold was set at 20 dB and the number of clusters at four for the non-dynamic case. As demonstrated in the graph, when using the “elbow” method, the algorithm performed better and was able to approach the target in approximately 10 to 20 s less, depending on the standard deviation of the additive noise. As a result, it was beneficial to include a heuristic method that determines the optimal number of clusters, rather than setting a constant number regardless of the number of available UAVs.

## 6. Conclusions

In this paper, a group of UAVs was tasked to trace an RF-emitting node, located at a distant unknown location. Based on a graph structure that represented the UAV network, a graph convolutional network architecture was used to determine clusters among the UAVs, according to their proximity to the RF source and their current positions. By ordering these clusters and removing those that appeared ineffective at approaching the target, the proposed algorithm was able to lead the remaining UAVs close to the RF source quickly, while saving energy by landing the UAVs of the removed clusters. Due to the dynamic number of UAVs, a heuristic method was utilised to determine the optimal number of clusters at each instant. Furthermore, the GCN model used a deep learning formulation to optimise the RSSI loss when performing the clusterisation process. Simulation experiments demonstrated the increased performance of this method, when compared to our previous deterministic approach, and indicated the applicability of deep learning architectures in localisation problems.

## Figures and Tables

**Figure 1 sensors-21-03936-f001:**
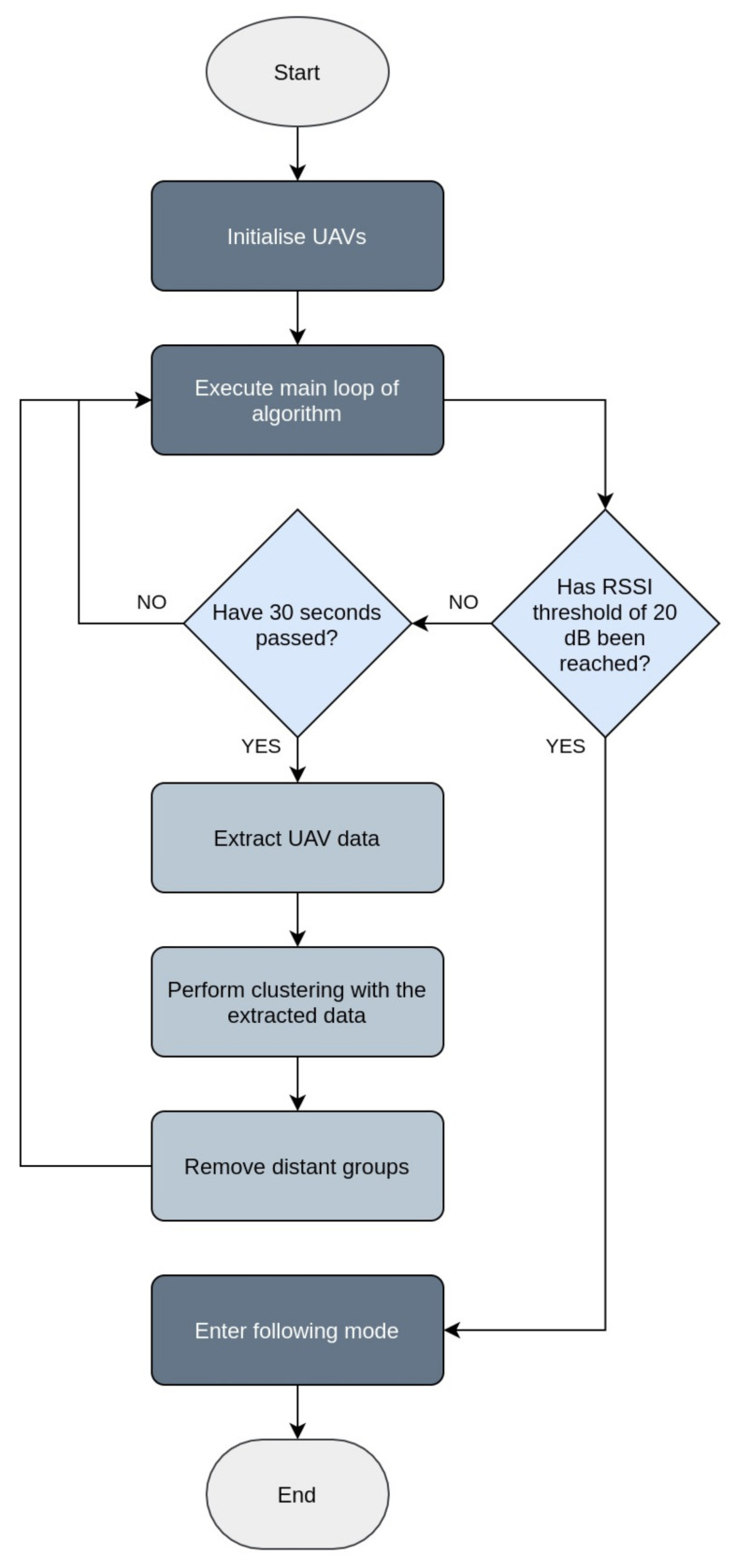
Flowchart demonstrating the proposed pipeline for approaching the mobile sensor.

**Figure 2 sensors-21-03936-f002:**
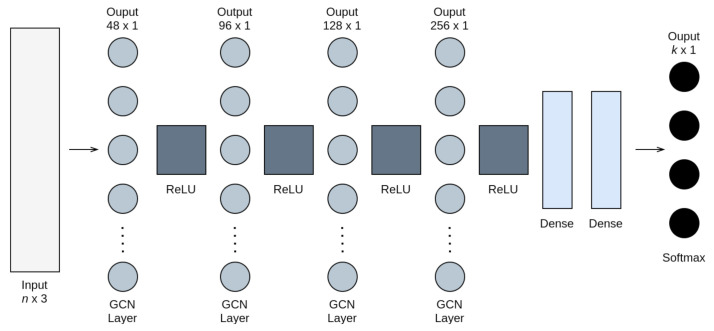
Structure of the deep learning network. The input layer’s dimensions are *n* × 3, where *n* is the number of UAVs in the UAV network. The output layer has *k* × 1 dimensions, with *k* being the number of clusters.

**Figure 3 sensors-21-03936-f003:**
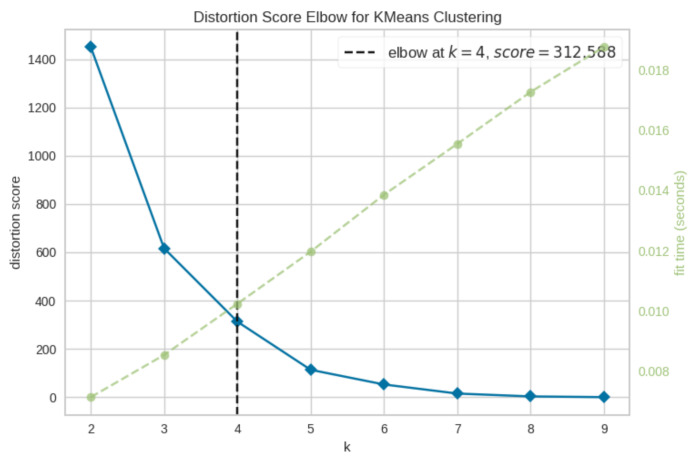
Example of a plot demonstrating the elbow method. The distortion score is computed as the sum of squared distances from each point to its assigned centre. In this case, the number of selected clusters should be 4.

**Figure 4 sensors-21-03936-f004:**
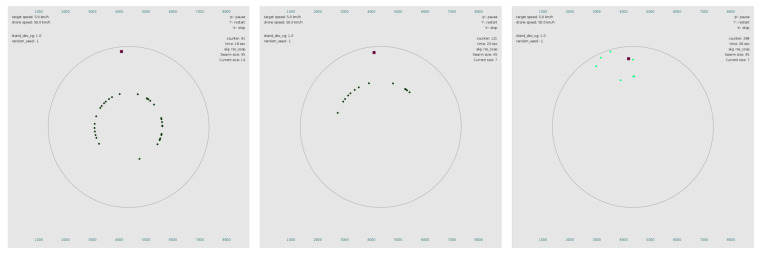
Simulator instances at different timestamps of the simulation. The red square represents the target sensor, and the dots indicate the UAVs. The left subfigure depicts the swarm 90 s after initiating the algorithm. The middle subfigure corresponds to 120 s, while the last subfigure demonstrates the tracing process after the threshold for terminating the DL procedure has been reached.

**Figure 5 sensors-21-03936-f005:**
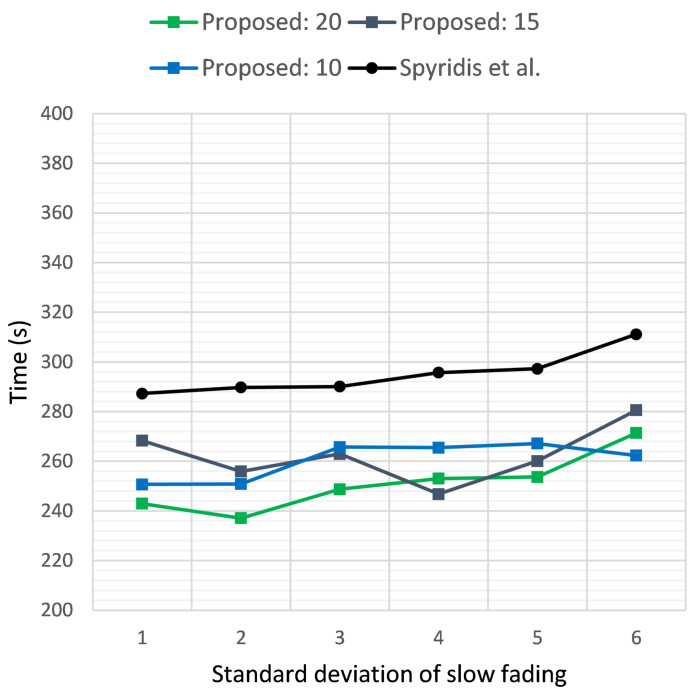
Time required for the first UAV to reach the target sensor when placed at a distance of 2 km, versus the standard deviation σ of the additive noise. Comparison of the proposed method at clustering termination thresholds of 20, 15 and 10 dB and the previous approach.

**Figure 6 sensors-21-03936-f006:**
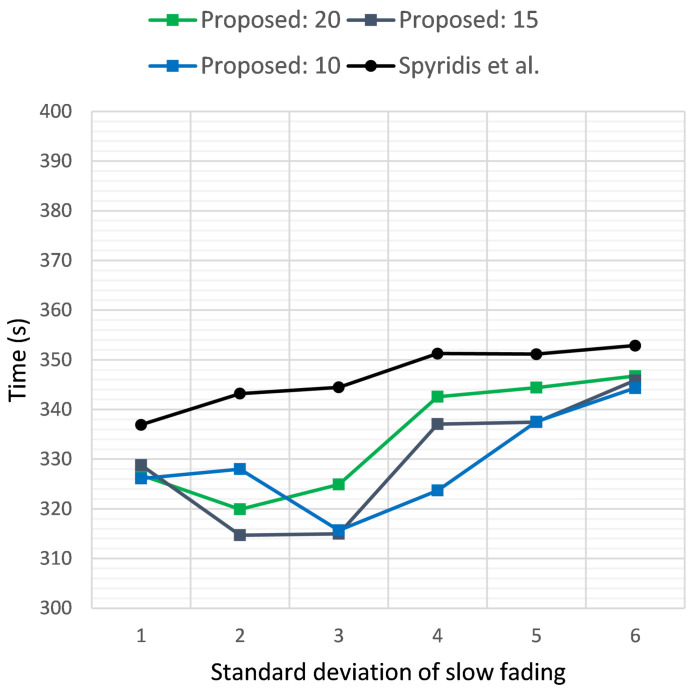
Time required for the first UAV to reach the target sensor when placed at a distance of 3 km, versus the standard deviation σ of the additive noise. Comparison of the proposed method at clustering termination thresholds of 20, 15 and 10 dB and the previous approach.

**Figure 7 sensors-21-03936-f007:**
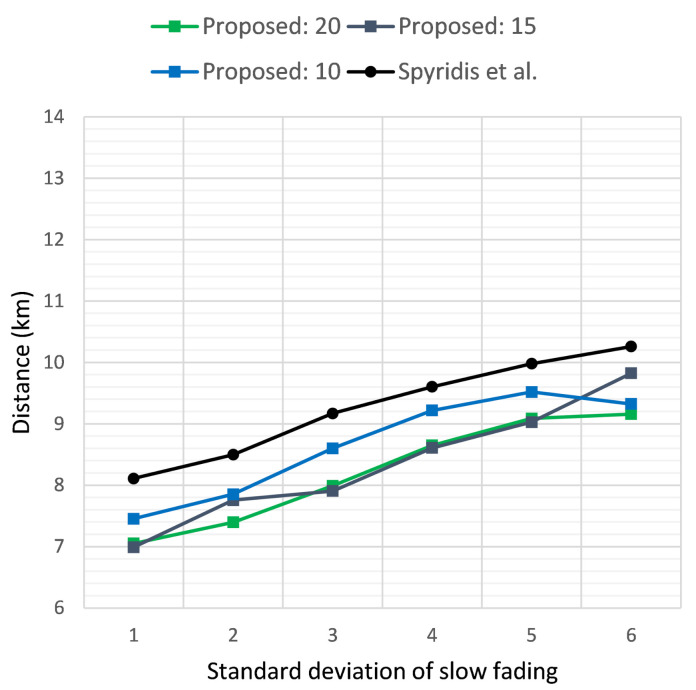
Average distance covered per UAV when the target sensor is placed at a distance of 2 km, versus the standard deviation σ of the additive noise. Comparison of the proposed method at clustering termination thresholds of 20, 15 and 10 dB and the previous approach.

**Figure 8 sensors-21-03936-f008:**
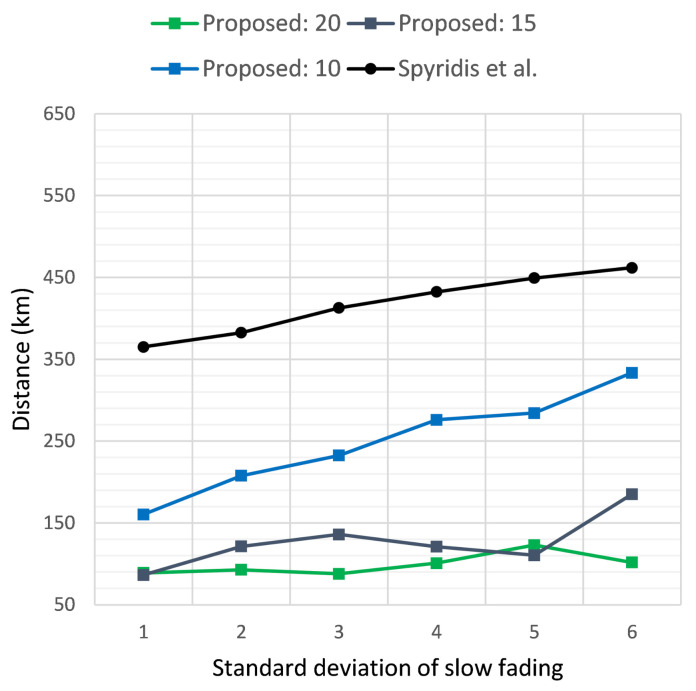
Total distance covered by all UAVs when the target sensor is placed at a distance of 2 km, versus the standard deviation σ of the additive noise. Comparison of the proposed method at clustering termination thresholds of 20, 15 and 10 dB and the previous approach.

**Figure 9 sensors-21-03936-f009:**
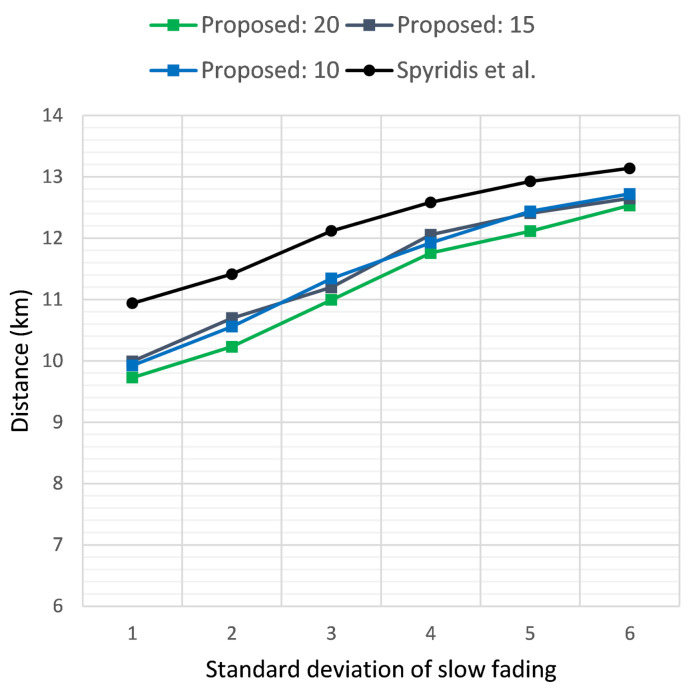
Average distance covered per UAV when the target sensor is placed at a distance of 3 km, versus the standard deviation σ of the additive noise. Comparison of the proposed method at clustering termination thresholds of 20, 15 and 10 dB and the previous approach.

**Figure 10 sensors-21-03936-f010:**
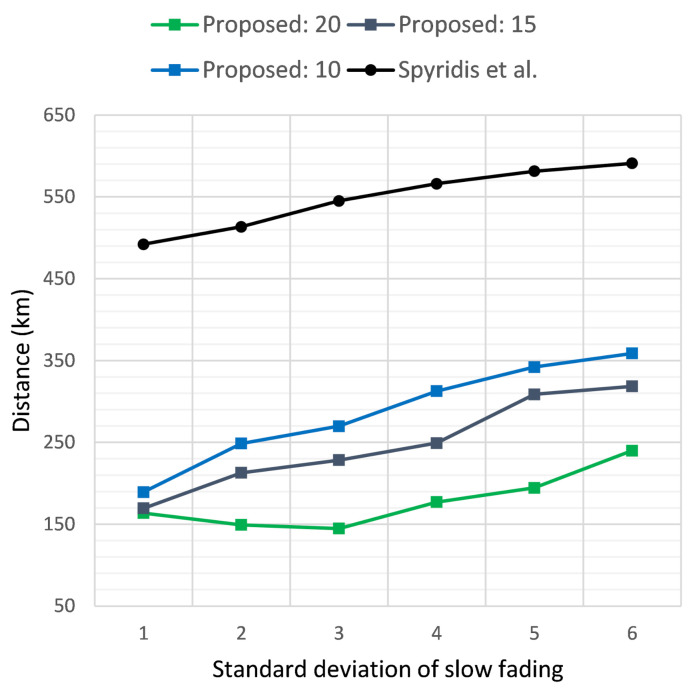
Total distance covered by all UAVs when the target sensor is placed at a distance of 3 km, versus the standard deviation σ of the additive noise. Comparison of the proposed method at clustering termination thresholds of 20, 15 and 10 dB and the previous approach.

**Figure 11 sensors-21-03936-f011:**
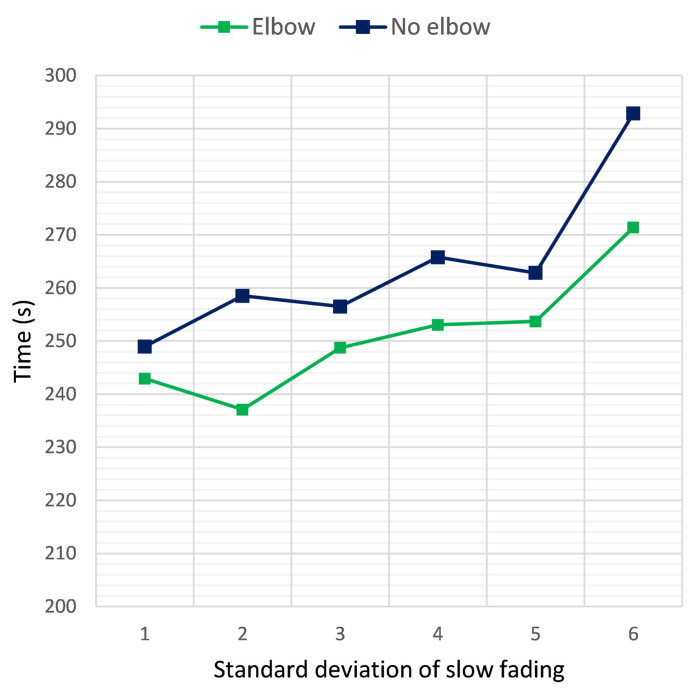
Time required for the first UAV to reach the target sensor when placed at a distance of 2 km, versus the standard deviation σ of the additive noise. Results at clustering termination thresholds of 20 dB, with and without using the elbow method to determine the number of clusters.

**Table 1 sensors-21-03936-t001:** Parameters of the simulation for each examined scenario. The two scenarios are differentiated in terms of the UAVs’ velocity and the target’s initial distance.

Parameter	Scenario 1	Scenario 2
Number of UAVs	45	45
Cluster update interval	30 s	30 s
UAV altitude	100 m	100 m
UAV velocity	40 km/h	50 km/h
Target velocity	5 km/h	5 km/h
Target distance	2 km	3 km
Target TX power	10 mW	10 mW
Target TX antenna gain	2 dBi	2 dBi
UAV RX antenna gain	2 dBi	2 dBi
Signal frequency	2400 MHz	2400 MHz
Total duration	1000 s	1000 s

**Table 2 sensors-21-03936-t002:** Average results for all values of the standard deviation σ of the additive noise for the DL method and the previous approach, at different thresholds. The target is placed at a 2 km distance. Time is measured in seconds and distance in meters.

Threshold	Proposed			Spyridis et al. [17]		
	Time	Average Distance	Total Distance	Time	Average Distance	Total Distance
10	260	8661	249,032	332	9201	414,214
15	268	8351	126,716	295	9270	417,322
20	251	8222	99,191	268	9345	420,705

## Data Availability

Not applicable.

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
