# Peer review of "Towards 6G IoT: Tracing Mobile Sensor Nodes with Deep Learning Clustering in UAV Networks"

_sensors, 2021, doi:10.3390/s21113936_

Round 1
Reviewer 1 Report
This work addresses a very interesting topic associated with 6G networks, which is the approximation of mobile devices by UAVs for providing telecommunication services. The use of a deep learning approach is a contemporary technique and sounds reasonable for the optimal automatic control of a swarm of drones. In general, the paper is well-written, and the results are promising, however, there are some points which needs to be taken care of, as explained below.
Section 2 discusses related work, that is other approaches in addressing the tracking problem, but it is not clear how the introduced technique is positioned (compared) within these approaches. A related paragraph at the end of the section is required.
There seems to be some inconsistency with the periodic interval of cluster formation. According to the algorithm described via a flowchart in Figure 1, clustering takes place every 10 seconds. However, according to Table 1, clustering in simulations takes place every 30 sec. This needs to be explained and corrected if needed. Also, some related discussion on the impact of this interval is recommended.
Also, the type of threshold implied in Figure 1 should be clearly stated in the figure.
In the first line of section 3.2, there is a typo “partitioning". Change it to partitioning.
An introductory paragraph in section 3 is recommended to be added. Starting a section directly with a subsection is not best practice. For instance, this paragraph could explain what kind of interconnections exist between which entities and how they are realistically implemented.
Some discussion on the reasons why the specific setup of the NN is proper (e.g., why that many layers) would be useful.
The simulation setup refers to X and Y coordinates. Why is the Z coordinated neglected? It is not clear, please explain.
Clarifications are required for considering only slow fading and not fast fading.
If the intention of Figure 4 is to have readable text, then there is a problem here. The fonts are too small.
The legends in the charts of subsection 5.1 are not quite explanatory. They should be explained in the caption. Also, there is some inconsistency in the way the authors refer to their technique. In the captions it is called “DL method”, while in the legend it is called “Proposed. Revise accordingly.
Typing mistake “approximately”.
The evaluation of the ‘elbow’ method for the determination of the number of clusters, as depicted in Figure 11, is interesting. However, it would be even more interesting to discuss how performance varies for varying number of clusters.
Author Response
Reviewer #1
- Comment:
This work addresses a very interesting topic associated with 6G networks, which is the approximation of mobile devices by UAVs for providing telecommunication services. The use of a deep learning approach is a contemporary technique and sounds reasonable for the optimal automatic control of a swarm of drones. In general, the paper is well-written, and the results are promising, however, there are some points which needs to be taken care of, as explained below.
Section 2 discusses related work, that is other approaches in addressing the tracking problem, but it is not clear how the introduced technique is positioned (compared) within these approaches. A related paragraph at the end of the section is required.
Response: Thank you very much for this insightful comment. In the revised manuscript we have included a related paragraph at the end of Section 2 to discuss where the introduced technique differs with other approaches:
“The method proposed in this work differs from hitherto studies, in that it aims to cover critical aspects associated with real-world scenarios. Towards this goal, it utilises simple RSSI sensors equipped on the UAVs, as opposed to more complex antennas, such as DOA or AOA sensors that have been proposed in other approaches. In addition, it incorporates the use of small and flexible multi rotor UAVs, in contrast to less versatile and cost-effective fixed-wing UAVs that are often used in similar works. When it comes to the tracking problem, the introduced method diverges from related studies, since it does not involve the conversion of the received signal’s power to distance from the target, which will inevitably have errors due to signal fluctuations. Finally, the deep learning model that is proposed does not require pre-training to a specific scenario to provide the solution, but operates in an unsupervised manner.”
- Comment:
There seems to be some inconsistency with the periodic interval of cluster formation. According to the algorithm described via a flowchart in Figure 1, clustering takes place every 10 seconds. However, according to Table 1, clustering in simulations takes place every 30 sec. This needs to be explained and corrected if needed. Also, some related discussion on the impact of this interval is recommended.
Response: Thank you for raising this issue. In the revised manuscript we have corrected the clustering interval to 30 seconds in all instances. We have also added text discussing the impact and this choice at the end of subsection 3.1:
“The optimal interval at which the clustering process occurs was examined through simulation evaluations, which determined that in general, lower intervals yield better performance. For the final implementation, the interval of 30 seconds was selected, since it provides an adequate performance, while being appropriate for use in a realistic scenario.“
- Comment:
Also, the type of threshold implied in Figure 1 should be clearly stated in the figure.
Response: Thank you for this valid point. In the revised manuscript we have updated Figure 1 to clearly indicate the threshold as well as the correct clustering interval.
- Comment:
In the first line of section 3.2, there is a typo “partitioning". Change it to partitioning.
Response: Thank you for raising this issue. In the revised manuscript we have corrected this and all typos accordingly.
- Comment:
An introductory paragraph in section 3 is recommended to be added. Starting a section directly with a subsection is not best practice. For instance, this paragraph could explain what kind of interconnections exist between which entities and how they are realistically implemented.
Response: Thank you for this insightful suggestion. In the revised manuscript we have added an introductory paragraph to Section 3 to introduce the reader to the problem, the involved entities and the method:
“In the considered problem, the target node refers to an IoT gateway which might be located on, or near an IoT device. The involved UAVs act as mobile relays, aiming to collect measurements or status updates from the target sensor. These measurements have as destination a 6G cellular base station, that will receive them though the assistance of the UAVs. The aim is to trace the IoT source node by exploiting wireless signal observations, through omni-directional RSSI sensors equipped in the UAVs.”
- Comment:
Some discussion on the reasons why the specific setup of the NN is proper (e.g., why that many layers) would be useful.
Response: Thank you for this valuable comment. In the revised manuscript we have added text discussing this in subsection 3.2 after presenting the NN layers:
“This specific setup allows for better feature representations and correlations among the input parameters in the considered clustering problem. The size of the network was selected according to the involved data, aiming to avoid over-fitting issues.“
- Comment:
The simulation setup refers to X and Y coordinates. Why is the Z coordinated neglected? It is not clear, please explain.
Response: Thank you for raising this valid point. The Z coordinate is neglected since it represents the altitude of the UAVs, which all fly at the same height. Therefore, this coordinate does not affect the clustering process. In the revised manuscript we have added text to Section 4 to explain this:
“The Z coordinate is not required, as it corresponds to the flying altitude, which is the same for all UAVs and therefore does not affect the clustering outcome.“
- Comment:
Clarifications are required for considering only slow fading and not fast fading.
Response: Thank you for raising this issue. Indeed we consider fast fading negligible since it mostly affects scenarios where close distances are involved, which is not the case in the considered problem. In the revised manuscript we have added text clarifying this in Section 4:
“The effect of fast fading was considered negligible since the involved distances are large and the UAVs fly in an open environment with not many obstacles that can result in multi-path propagation.”
- Comment:
If the intention of Figure 4 is to have readable text, then there is a problem here. The fonts are too small.
Response: Thank you for raising this valid point. The purpose of Figure 4 is mainly to offer a visual presentation of our simulator for demonstration purposes. Unfortunately, it is not feasible to increase the fonts to readable levels for inclusion in the manuscript without obscuring the involved entities. The simulator can provide a lot of information for display according to the experiment requirements, which is clearly visible when running it in full-screen.
In the experiment demonstrated in Figure 4, it displays the target sensor and UAV speeds, as well as the noise level and the current random seed on the left side. On the right side, it displays information about controls, such as pause, restart and stop, while below it displays the current time of the simulation, the number of total and current UAVs and finally, the name of the current algorithm.
- Comment:
The legends in the charts of subsection 5.1 are not quite explanatory. They should be explained in the caption. Also, there is some inconsistency in the way the authors refer to their technique. In the captions it is called “DL method”, while in the legend it is called “Proposed. Revise accordingly.
Response: Thank you for this insightful comment. In the revised manuscript we have updated the captions to be consistent, and explained the information shown in the legends.
- Comment:
Typing mistake “approximately”.
Response: Thank you for raising this issue. In the revised manuscript we have corrected all typing mistakes.
- Comment:
The evaluation of the ‘elbow’ method for the determination of the number of clusters, as depicted in Figure 11, is interesting. However, it would be even more interesting to discuss how performance varies for varying number of clusters.
Response: Thank you for this insightful comment. In the revised manuscript we have added a paragraph in subsection 3.2.3 to discuss this issue:
“The performance of the proposed algorithm varies for different number of clusters, but because in each clustering interval the setup is unique for every experiment, it is intrinsically difficult to deduce a universal correlation. For this this reason, the "elbow" method appears exceptionally useful, since it helps to automate the whole process and determine the appropriate number of clusters. Even though in certain cases it might yield a suboptimal solution, in most occasions the result is optimal.”
Reviewer 2 Report
This paper propose a deep learning model for UVA cluster based on a graph conventional network (GCN) to approach the target as quickly as possible. The paper is well written and easy to follow. The reviewer has the following concerns:
- In figure 1 for the Flowchart demonstrating the proposed pipeline for approaching the mobile sensor, please given the reason for choose 10 seconds, has this time is optimized by simulation or selected random?
- In figure 3, the definition for distortion score should be given.
- In equation (1), the meaning for d and f should be explained.
Author Response
Reviewer #2
- Comment:
This paper propose a deep learning model for UVA cluster based on a graph conventional network (GCN) to approach the target as quickly as possible. The paper is well written and easy to follow. The reviewer has the following concerns:
In figure 1 for the Flowchart demonstrating the proposed pipeline for approaching the mobile sensor, please given the reason for choose 10 seconds, has this time is optimized by simulation or selected random?
Response: Thank you very much for this insightful comment. In fact, the final clustering interval was selected at 30 seconds, as was written in the original text. In the revised manuscript we have updated Figure 1 with the correct number and added text discussing this in subsection 3.1:
“The optimal interval at which the clustering process occurs was examined through simulation evaluations, which determined that in general, lower intervals yield better performance. For the final implementation, the interval of 30 seconds was selected, since it provides an adequate performance, while being appropriate for use in a realistic scenario.“
- Comment:
In figure 3, the definition for distortion score should be given.
Response: Thank you for this valid point. In the revised manuscript we have updated the caption of Figure 3 to provide the definition of the distortion score:
“The distortion score is computed as the sum of squared distances from each point to its assigned centre.“
- Comment:
In equation (1), the meaning for d and f should be explained.
Response: Thank you for this valuable suggestion. In the revised manuscript we have added text explaining the variables d and f below the equation:
“... , where d is the distance between transceiver and receiver, f is the signal frequency, ...”